# Divergence between Hemichannel and Gap Junction Permeabilities of Connexin 30 and 26

**DOI:** 10.3390/life13020390

**Published:** 2023-01-31

**Authors:** Ji Xu, Bruce J. Nicholson

**Affiliations:** 1Department of Pharmacology, Institute of Neuroscience, School of Basic Medical Sciences, Zhengzhou University, Zhengzhou 450001, China; 2Department of Biochemistry and Structural Biology, University of Texas Health at San Antonio, San Antonio, TX 78229, USA

**Keywords:** connexin, hemichannels, gap junctions, permeability, ATP

## Abstract

Cx30 has been proposed to play physiological functions in the kidney and cochlea, and this has often been associated with its hemichannel role (deafness mutants frequently affecting hemichannels more than gap junctions), implicated in ATP release. Here, we used heterologous expression systems (*Xenopus oocytes* and N2A cells) to describe the properties of Cx30 hemichannels, with the objective of better understanding their physiological functions. As previously observed, Cx30 hemichannels gated in response to transmembrane voltage (V_0_) and extracellular [Ca^2+^] (pK[Ca^2+^] of 1.9 μM in the absence of Mg^++^). They show minimal charge selectivity with respect to small ions (ratio of Na^+^: K^+^: Cl^−^ of 1: 0.4: 0.6) and an MW cut-off for Alexa Dyes between 643 (Alex 488) and 820 Da (Alexa 594). However, while cations follow the expected drop in conductance with size (Na^+^ to TEA^+^ is 1: 0.3), anions showed an increase, with a ratio of Cl^−^ to gluconate conductance of 1:1.4, suggesting favorable interactions between larger anions and the pore. This was further explored by comparing the permeabilities of both hemichannels and gap junctions to the natural anion (ATP), the release of which has been implicated in Ca^++^ signaling through hemichannels. We extended this analysis to two closely related connexins co-expressed in the cochlear, Cx26 and Cx30. Cx30 and 26 hemichannels displayed similar permeabilities to ATP, but surprisingly Cx26 gap junctions were six times more permeable than their hemichannels and four times more permeable than Cx30 gap junctions. This suggests a significant physiological difference in the functions of Cx26 and Cx30 gap junctions in organs where they are co-expressed, at least with regard to the distribution of energy resources of the cells. It also demonstrates that the permeability characteristics of hemichannels can significantly diverge from that of their gap junctions for some connexins but not others.

## 1. Introduction

Connexin hemichannels have been suggested to play various physiological functions by releasing large molecules from cells, such as prostaglandins [1], NAD^+^ [2] and ATP [3]. Cx30 proteins are widely distributed in the brain, skin, cochlea and kidney and are frequently co-expressed with the closely related Cx26. The Cx30 hemichannels have been suggested to play physiological functions in several of these tissues, particularly the ear, where mutants that selectively affect Cx30 or Cx26 hemichannels have been associated with disease [4]. Immunostaining of Cx30 in the kidney cortex revealed that Cx30 is distributed in the apical membrane of some cortex collecting duct (CCD) epithelial cells [5]. Since these channels are not distributed on the junctional membrane between neighboring cells, they cannot contribute to gap junction channels but could form open hemichannels. This was consistent with the demonstration that epithelial cells release ATP after mechanical stimulation, a capability that was lost in Cx30 knockout animals [6], along with the ability to propagate Ca^2+^ signals [7]. In the cochlea, Ca^2+^ wave propagation between supporting cells has also been associated with Cx30 hemichannels. Ca^2+^ waves, induced by puffing ATP from a pipette to initiate [Ca^2+^]_i_ increase in several cells, are significantly reduced in Cx30 knockout animals [8]. An extracellular ATP biosensor was also used in this study to demonstrate that ATP is released from supporting cells in response to uncaging intracellular InsP_3_, a response that is abolished in Cx30 knockout animals. The relative contributions of Cx30 and 26 to Ca^2+^ wave propagation by either extracellular routes (as suggested above) and direct intercellular transfer of IP_3_ through gap junctions is comprehensively assessed in [9].

While the above results all support signaling that is mediated by ATP release through Cx30 hemichannels, other studies have shown that Cx30 gap junction channels are impermeable to similarly sized and negatively charged fluorescent dyes, such as Lucifer Yellow, while being readily permeable to positively charged molecules of similar sizes, such as neurobiotin [10,11]. The only study to directly assess the permeability characteristics of Cx30 hemichannels demonstrated much greater permeability for neutral glucose than for anionic ATP and glutamate, consistent with the cationic preference observed for Cx30 gap junctions [12]. However, it should be noted that mutant Cx30 hemichannels associated with skin disease and hearing loss have been inferred to be quite permeable to ATP [13].

It has been broadly assumed in the literature that hemichannels and gap junctions composed of the same connexin should share similar permeability characteristics. This has been carefully established only in the case of Cx46, where direct comparisons of the electrical properties of hemichannels and gap junctions were highly consistent [14]. Other studies have observed significant differences between hemichannels and gap junctions of the same connexin, perhaps the most extreme being Cx43, where hemichannels, in contrast to gap junctions, passed several larger permeants in the absence of measurable ionic conductance [15]. These latter observations are consistent with the recent hemichannel structure of Cx31.1 [16], which shows significant differences with published gap junction structures, albeit of different connexins [17,18,19].

Could differences in hemichannel and gap junction permeabilities reconcile the proposed role of Cx30 hemichannels in the release of ATP in the ear, with the demonstrated preference of its gap junction channels for cationic permeants? Or could this be explained by confounding activities of the other major connexin expressed in the ear, Cx26? While Cx26 and Cx30 are closely related in sequence, the permeabilities of their gap junction channels diverge, with Cx26 favoring anions [20] and Cx30 favoring comparatively larger cations [10,11]. To extend these analyses to a comparison of both hemichannels and gap junctions composed of these two connexins, we have measured their permeabilities to larger permeants, including ATP itself, normalized to conductance. This has revealed interesting differences that support the concept that hemichannels may have distinct properties to their corresponding gap junction channels and indicate that the two connexins may play distinct physiological roles.

## 2. Materials, Methods and Analyses

### 2.1. DNA Constructs Preparation

The plasmid of human Cx30 (hCx30) in the pIRESneo3 vector was kindly provided by Dr. Sabrina W. Yum (Children’s Hospital of Philadelphia). The hCx30 insert was excised with EcoRV and NotI and then subcloned into the pBSCMXT vector. Then, hCx30-pBSCMXT was linearized by AccI, and cRNAs were prepared using a mMessage mMachine kit (Ambion, Austin, TX, USA). The resultant cRNAs were quantitated after DNase treatment using an O.D. 260 nm measurement. For expression in N2A cells, the hCx30 insert was excised from the original vector with EcroRV and BamHI and subcloned into the pIRES-EGFP vector.

### 2.2. Xenopus oocytes Preparation

Oocytes were taken from ovarian lobe tissue and surgically removed from adult female *Xenopus laevis* after anesthesia in 0.2% tricaine chilled to 4–6 °C. The tissue was then separated into smaller segments containing about 6–12 oocytes and incubated at 22 °C for 20 min in OR2 buffer (83 mM NaCl, 2 mM KCl, 1 mM MgCl_2_ and 5 mM HEPES at pH 7.4 with NaOH) containing 1.5 mg/mL collagenase. After washing with OR2, stage V and VI oocytes were separated and manually defolliculated as necessary. Oocytes were continuously bathed at 17 °C in half-strength L15 media (Sigma, St. Louis, MO, USA) supplemented with penicillin (0.3 mg/mL), gentamicin (0.3 mg/mL) and streptomycin (0.3 mg/mL).

### 2.3. Recording Hemichannel Current in Xenopus oocytes

Oocytes were co-injected with 20 ng of hCx30 RNA mixed with 8 ng of antisense oligonucleotide directed against nucleotides −5–25 of endogenous *Xenopus* Cx38. Approximately 48 h after injection, hemichannel currents were recorded with a two-electrode voltage clamp with Geneclamp500 Amplifiers (Axon Instruments, La Jolla, CA, USA). Clamping electrodes were prepared from capillary glass (1B150F-4, World Precision Instruments, Sarasota, FL, USA) using a horizontal micropipette puller (model P-87, Sutter Instrument). Electrodes with resistance ~1 MΩ when filled with 150 mM KCl were used. Oocytes were bathed in ND96 solution (96 mM NaCl, 2 mM KCl, 1 mM MgCl_2_ and 5 mM HEPES at pH 7.4) for recording. The Ca^2+^ concentration was controlled by adding CaCl_2_ and EGTA. The final Ca^2+^ concentration was calculated by Maxchelator (WEBMAXC Standard version, Stanford University). Data were acquired and analyzed using Pclamp 8 software (Axon Instruments, La Jolla, CA, USA).

### 2.4. Whole Cell Recording in N2A Cells

N2A cells, kindly provided by Dr. Jean Jiang (The University of Texas Health at San Antonio), were maintained in MEM medium (Mediatech, Manassas, VA, USA) supplemented with 10% fetal bovine serum at 37 °C in a CO_2_ incubator (5% CO_2_–95% ambient air). One day after seeding onto glass coverslips, N2A cells were transfected by Lipofectamine LTX (Thermo-Fisher, Waltham, MA, USA) according to the manufacturer’s recommended protocol. Electrophysiological experiments were performed 1–3 days after transfection on cells that showed GFP signal in a chamber containing 130 mM NaCl, 10 mM KCl, 1 mM MgCl_2_, 5 mM HEPES and 5 mM glucose with pH 7.4 at room temperature (∼21 °C) mounted on a Zeiss Axiovert 10 fluorescent microscope (Carl Zeiss. Dubln, CA, USA). Patch pipettes of 1–2 MΩ resistance were pulled from glass capillaries with a horizontal puller (P-87, Sutter Instruments–Sutter Instruments, Novato, CA, USA) and filled with 140 mM KCl, 1 mM CaCl_2_, 1 mM MgCl_2_, 5 mM EGTA and 5 mM HEPES at pH 7.2. Transmembrane voltage was clamped, and the current was recorded by a PC-505 Warner Instruments amplifier (Warner Instruments Inc., Hamden, CT, USA).

### 2.5. Constant Field Equations to Calculate Relative Ion Permeabilities

The Goldman–Hodgkin–Katz (GHK) constant field equation described below was used to calculate the relative ion permeabilities according to the measured reversal potentials (*V*_rev_):(1)$$V\mathrm{rev}=\frac{RT}{F}\ln(\frac{\sum P_{C}{\left[C\right]}_{o}+\sum P_{A}{\left[A\right]}_{i}}{\sum P_{C}{\left[C\right]}_{I}+\sum P_{A}{\left[A\right]}_{o}}),$$
where the [C] and [A] denote cation and anion concentration, respectively, and the subscripts i and o denote intracellular and extracellular, respectively. *P*_C_ and *P*_A_ are the cation and anion permeabilities, respectively, and *R*, *T* and *F* have their usual meaning.

### 2.6. Dye Release and ATP Release Assay from Xenopus oocytes

Two days after injection of 20 ng connexin RNA, the whole cell membrane conductance of control and Cx26- or Cx30-expressing oocytes were measured by a two-electrode voltage clamp. As we had demonstrated that Cx26 and Cx30 hemichannel conductance could reach a steady state after 10 min equilibrium in Ca^2+^-free KCl solution (not shown), the conductances were measured 10 min after transferring from L15 medium to KCl solution. After conductance measurement, oocytes were injected with 32 nL of either 10 mM Alexa dyes or 1.25 mCi/mL ^35^S-labeled 10 μM ATP-γ-S (Perkin-Elmer, Austin, TX, USA) in L15 medium and then washed three times with L15 medium and Ca^2+^-free KCl solution once (85 mM KCl, 1 mM MgCl_2_ and 5 mM HEPES at pH 7.4). They were then transferred into Eppendorf tubes with 110 μL KCl, Ca^2+^-free solution. After 30 min, 100 μL medium was collected and mixed with 900 μL water. The remaining oocyte was rinsed once with 110 uL KCl and then lysed in 0.1% SDS lysis buffer. Alex dyes released to the media (M) were measured at the appropriate wavelength on a Fluromax-3 fluorimeter (Horiba, Japan), and ATP released to the media (M) was measured by scintillation counting on an LS6500 Beckman instrument (Beckman Coulter Life Sciences, Indianapolis, IN, USA) after mixing the samples with 10 mL UniverSol scintillation emulsifier (MP Biomedicals LLC, Irvine, CA, USA). The lysed oocyte (O) was diluted to the same volume and similarly measured in order to generate the M/O (media/oocyte) ratio of fluorescence or radiation.

### 2.7. ATP Transfer Assay in Xenopus oocyte Pairs

One day after injection of 5 ng Cx26 or Cx30 cRNA, oocytes were stripped off their vitelline membrane and paired with each other in agar wells. The next day, transjunctional conductances were measured by a dual-cell voltage clamp [21] before each ATP transfer experiment. Initial conductances were required to be between 5 and 50 μS so as to ensure a significant signal above the background while avoiding the possibility that cytoplasmic bridges could have formed between the oocytes.

Immediately following conductance measurements, one oocyte is injected with 32 nL of 1.25 mCi/mL ^35^S-labeled ATP-γ-S (Perkin-Elmer). Precisely 60 min after injection, the acceptor and donor cells are separated by micro-dissection under a Stereo 10X microscope. The acceptor and donor cells are immediately removed separately from the agar well and lysed with a 0.1% SDS buffer, and each is placed in 20 mL scintillation vials (Research Products International, Mount Prospect, IL, USA). A total of 10 mL of UniverSol scintillation cocktail is added (MP Biomedical), and the vials are evenly shaken prior to scintillation counting using a Beckman Coulter LS6500 scintillation counter (Beckman Coulter Life Sciences, Indianapolis, IN, USA). Quantitative measurements of Alexa transfer through both Cx30 and 26 channels had been previously performed [22].

## 3. Results

### 3.1. Cx30 Forms Voltage- and Ca^++^-Sensitive Hemichannels in Xenopus oocytes and N2A Cells

We initially measured the properties of Cx30 hemichannels expressed in *Xenopus oocytes*, co-injected with antisense oligonucleotides to endogenous Xe Cx38. Expression was confirmed with immunofluorescence, which showed general cytoplasmic expression with a concentration at the membrane (Figure 1). The two-electrode voltage clamp of *Xenopus oocytes* in a Ca^2+^-free solution revealed much larger whole-cell currents (Figure 2B) than control cells (Figure 2A). Initial currents linearly increased with applied membrane voltage (Figure 2C), but currents decayed on an applied voltage of either polarity (Figure 2B). At negative polarities, steady-state currents could be fitted with a Boltzman with a V_0_ of ~−35 mV (Figure 2D). Very similar results were obtained by expressing Cx30 in N2A cells, which express minimal levels of endogenous connexins (cf. Figure 1 and Figure 2E,F), indicating a small effect of the expression system. Initial currents were linear with voltage, with a slight increase seen at negative potentials > −80 mV (Figure 2G). Steady-state conductances also dropped, consistent with a Boltzman decay but with a somewhat larger V_0_ than was seen in oocytes (>−60 mV). The physiological basis of this one difference was not explored further here.

Another measure of Cx30 hemichannel expression was the drop in membrane potential in Cx30 oocytes in a Ca^++^-free solution where hemichannels have been shown to open. This was not seen in control nor Cx30 oocytes in 1.8 mm Ca^++^ (Figure 3A). Detailed characterization of the sensitivity of Cx30 hemichannels to extracellular Ca^2+^ was tested using EGTA to buffer extracellular [Ca^2+^] at different concentrations. Since extracellular Mg^2+^ cannot be depleted from the extracellular medium, as it is required to maintain the membrane conductance at a level that can be controlled by voltage clamp, only EGTA (which is more Ca^2+^ specific) was used. Plotting normalized conductance (to the maximum conductance of the specific cell) against the log of extracellular [Ca^2+^] displays a sigmoid relationship, with pK[Ca^2+^] of 1.9 µM (Figure 3B). Therefore, at the extracellular [Ca^2+^] used in control experiments (1.8 mM), the P_o_ of Cx30 hemichannels should be lower than 0.1.

### 3.2. Permeability of Cx30 Hemichannels to Ions

To assess the permeability characteristics of Cx30 hemichannels, we first determined their ion selectivity by plotting the I–V relationship and reversal potential at different extracellular NaCl concentrations. A 0.5 s ramp protocol was used to minimize the effect of time-dependent voltage gating. Current traces recorded in 24, 120 and 240 mM NaCl intersect in the first quadrant, with the reversal potential (V_rev_—X-intercept) shifting to more positive values with higher concentrations of NaCl (Figure 4 and Table 1). According to the interpretation developed in [23], the ratio of permeabilities to Na^+^, K^+^ and Cl^−^ is 1: 0.38: 0.63, as calculated by the GHK equation. Note that this general lack of significant cation or anion selectivity is distinct from the endogenous *Xenopus* Cx38 channels, which show permeability ratios of 1:1:0.24, indicating a significant cation selectivity.

Permeability to larger ions was tested through the replacement of extracellular Na^+^ with TEA^+^ or Cl^−^ with gluconate while keeping other ionic compositions constant. Table 2 shows the reversal potentials measured at different percentages of substitution. Based on these measurements, the permeability ratio of Na^+^ to TEA^+^ is 1: 0.25, close to that predicted from their relative size and diffusion rates. However, the anionic permeabilities of Cl^−^ to gluconate were 1: 1.42, with the larger ion showing a higher permeation than the smaller one, suggesting that the pore may contain weak binding sites for gluconate that could enhance its flux through the channel. This is similar to what was observed for some gap junctions’ permeabilities to fluorescent Alexa dyes [22], which exceeded that predicted from their conductance unless attractive forces between the permeant and channel lining were included.

### 3.3. Permeability of Cx26 and Cx30 Hemichannels to Fluorescent Dyes and ATP

To establish permeability limits for the Cx30 channel, we used a series of Alexa dyes, as has been performed for several other connexins [22], including the closely related Cx26, which we used as a comparison here. Expression of Cx26 in oocytes was electrophysiologically confirmed, with gap junction intercellular currents showing the characteristic voltage dependence previously published (cf. Appendix A to [21]). The leak from XeCx38 antisense-oligonucleotide- (oligo) injected oocytes was small, although, as previously reported, it was greatest for the smaller Alexa 350 dye (Weber et al., 2004). After incubating the oocytes in Ca^2+^-free solution for 60 min, the medium collected from Cx26- or Cx30-expressing oocytes had significantly more Alexa 350 (~3 fold) and 488 (~2 fold), but not Alex 594, than that collected from oocytes injected with oligo alone (Figure 5A). Therefore, both Cx26 and Cx30 hemichannels appear to have a cut-off limit between MW 650 (Alexa 488) and 800 (Alexa 594). The same exclusion limits have also been reported for Cx26 gap junctions [10,22]. Thus, with respect to size exclusion limits for anionic dyes, Cx26 hemichannels and gap junctions have similar properties. However, in contrast to what we observe here, Cx30 gap junctions had been found to be impermeable to either Alexa 488 (or similarly sized and charged Lucifer Yellow) or Alexa 594 [10], suggesting that Cx30 gap junctions have lower exclusion limits than their hemichannels, at least for anionic permeants.

Since hemichannels have been suggested to mediate physiological functions by releasing ATP, we also directly measured the permeability of Cx30 and 26 hemichannels to ATP by tracking the release of the ^35^S-labeled, non-hydrolyzable ATP-γ-S analog injected into oocytes. After incubation in Ca^2+^-free solution for 60 min, the medium collected from both Cx30- and Cx26-expressing oocytes contained 2–2.5 fold the counts of the media from uninjected oocytes, demonstrating that both hemichannels are permeable to ATP (Figure 5B). The relative permeability compared to uninjected oocytes was similar to that seen with Alexa 488 and Alexa 350, indicating no significant differences in permeability between these larger anions.

### 3.4. Comparison of Cx26 and Cx30 Hemichannel Permeability to ATP-γ-S

To obtain a more specific measure of permeability, we directly compared ATP release from Cx26- and Cx30-expressing oocytes to their membrane conductance. As shown in Figure 5B, there was a consistent and reproducible background of ATP leak from oligo-injected oocytes, and this was subtracted from all release measurements. Similarly, the membrane conductance of oligo-injected oocytes, reflecting other channels or background from XeCx38, was low and very consistent between all tested oocytes and was also subtracted from all conductance measurements of connexin-expressing oocytes. After these adjustments, ATP release was plotted against connexin-dependent membrane conductance for each cell, yielding linear plots for both Cx30 and 26 hemichannels (Figure 6A), with high correlation coefficients (0.86 and 0.97, respectively). The slopes of these plots, reflecting the permeability of Cx26 and Cx30 hemichannels for ATP, were not significantly different.

### 3.5. Comparisons of Gap Junction and Hemichannel Permeabilities of Cx26 and 30

Given that the lack of significant differences we observe in permeability between Cx26 and 30 hemichannels to larger anions (Alexa dyes and ATP) are at odds with reports of the gap junction permeabilities of these same connexins to anionic dyes [10,11], we expanded our comparisons of ATP permeability to Cx26 and Cx30 gap junctions. This was conducted in a comparable manner to how we assessed hemichannel permeability, but in this case, the transfer of radiolabeled ATP from an injected “donor” to an “acceptor” cell was measured by manually separating and counting each cell. Background values, which were obtained by measuring radionucleotide transfer to acceptor cells in oocyte pairs that were only injected with antisense oligos to endogenous XeCx38, were minimal and subtracted from the transfer seen between oocyte pairs expressing Cx26 or Cx30. Since the junctional conductance between all pairs was measured prior to ATP injection, ATP transfer could be plotted against conductance as it was for hemichannel measurements, yielding similarly high correlation coefficients (0.89 for Cx30 and 0.93 for Cx26). In the case of gap junctions, however, Cx26 showed a four-fold higher permeability for ATP than Cx30 gap junction channels (Figure 6B).

The measurements of hemichannel and gap junction permeability were performed in a very similar fashion, as the passage of ATP to the extracellular space or an adjacent cell at a standardized concentration gradient (based on constant ATP injections) and normalized to the membrane or junctional conductance electrophysiologically measured. The gap junction channel and hemichannel currents should be comparable by measuring hemichannel conductances in a KCl bath solution, also assumed to be the major intracellular charge carriers. This allows the data for hemichannels and gap junctions to be directly compared, demonstrating that Cx30 hemichannels and gap junctions have similar permeabilities to ATP (Figure 6C). In contrast, Cx26 hemichannels are ~six times more permeable to ATP than Cx26 hemichannels (Figure 6D). This certainly provides direct evidence that, depending on connexin composition, the permeabilities of gap junctions and hemichannels can be either comparable (as for Cx30) or highly divergent (as for Cx26).

## 4. Discussion

### 4.1. Permeability of Connexin Hemichannels to Large Molecules

Connexin hemichannels have been increasingly implicated in diseases, mostly thought to be associated with their excessive openings that can lead to cell death [24,25]. However, they have also been implicated in physiologically relevant roles in processes that include inflammatory responses [26,27], osteogenesis [28] and Ca^++^ wave propagation [8]. These responses are thought to be mediated by the release of metabolites, such as ATP and PGE. To fully understand their roles, however, we need to better understand not only what controls their opening but the permeability characteristics of hemichannels comprised of different connexins. To date, such permeability studies have focused more on gap junction channels, with the general assumption being that this would also reflect hemichannels properties. However, in the cases where direct comparisons have been made, only a few have gap junction and hemichannel permeabilities have been comparable (i.e., Cx46—[29]), while in others, they have seemed very distinct (e.g., Cx43 [12,30]).

To pursue this comparison, we chose two of the most closely related connexins, Cx26 and 30, to assess the degree to which their gap junction and hemichannel properties diverge. This is also physiologically relevant, as they are often co-expressed, with both Cx30 and 26 gap junctions and hemichannels being implicated in cochlear function, as evidenced by the number of mutations in these genes that are associated with deafness. ATP release through Cx30 hemichannels has been proposed to play a role in the ear [6,8] and has also been implicated in renal function [6]. However, little has been performed to analyze the permeability of Cx30 or Cx26 hemichannels for molecules other than small ions, except for one semi-quantitative comparison of Cx30 and 43 hemichannels [12], which showed that glucose, glutamate and ATP (in descending order of permeation) pass Cx30, but not Cx43, hemichannels. Glutathione and lactate failed to pass through either channel. However, no normalization to channel number was done, and other papers have concluded that Cx43 hemichannels are permeable to some of these same metabolites.

Here we have used ion replacement experiments to investigate Cx30 and 26 channel properties, along with direct measures of Alexa dye and ATP release in *Xenopus oocytes* to assess permeability for larger species quantitively. The latter established the cut-off size for Cx30 and Cx26 hemichannels as between 643 and 820 Da (at least for anions), which is comparable to what was reported for their gap junction channels (at least in the case of Cx26 (22)). This is well above the molecular weight of all the metabolites tested in [30], including ATP (which showed limited permeability), lactate and glutathione (which showed no detectable permeation), suggesting that factors other than size need to be considered.

Ion replacement studies on Cx30 hemichannels showed the expected relative permeabilities of small (K^+^) and large (TEA) cations but surprisingly showed that Cx30 hemichannels are more permeable to gluconate than chloride ions. One possible explanation for the greater permeability of gluconate (despite its larger size and slower diffusion constant in a bulk solution compared with Cl^−^) would be affinity with the pore of Cx30 hemichannels. We have previously used the Poisson–Nernst–Planck modeling of flux through gap junctions to demonstrate that pore affinities for fluorescent dyes can account for permeabilities at two orders of magnitudes larger than those predicted by a simple diffusion model [22]. However, this preference of Cx30 hemichannels for larger anions contrasts with prior studies of Cx30 gap junctions that indicated they strongly favor cationic permeants (10,11), possibly indicating a divergence between permeability characteristics of Cx30 hemichannels and gap junctions.

### 4.2. Is a Hemichannel Just Half of a Gap Junction Channel?

Several hemichannels have been shown to have double the single-channel conductance of gap junction channels, as would be expected from the conductance pathway being half the length. This is true for Cx30, where hemichannel and gap junction conductances have been measured as 280 and 140–180 pS, respectively [31,32]. Cx26 shows the same general pattern, although significantly deviating from the 2:1 ratio (hemichannels: 340 pS [33]; gap junction: 135 pS [20]). However, given the relatively large pore size of hemichannels and gap junction channels, electrical conductance (i.e., the passage of small ions) may not tell us much about the selective properties of the pore. As noted above, the pore-lining may have an affinity for larger permeants, so the quantitative study of these may better elucidate physiologically relevant properties of the pore.

Comparisons of Cx26 and Cx30 hemichannel and gap junction channel permeabilities to ATP, normalized to ionic conductance, showed that Cx30 gap junctions and hemichannels had indistinguishable permeabilities for ATP. In sharp contrast, Cx26 gap junction channels were shown to have a six-fold higher permeability to ATP than its hemichannels. While we did not conduct a side-by-side comparison of Cx26 gap junctions and hemichannels for permeability to the Alexa dye series, the patterns observed for Cx26 hemichannels appear to parallel those reported for gap junctions, suggesting that the enhanced permeation of Cx26 gap junctions may be specific to ATP. Therefore, depending on the connexin composition and the permeant being studied, hemichannel properties can reflect those of the gap junction or significantly diverge.

Several factors could contribute to hemichannels and gap junction channels adopting different conformations that would differentially affect their permeability preferences:(1)They reside in different lipid environments, as gap junction plaques have specialized lipid contents resembling lipid rafts [34] that are unlikely to be preserved around unclustered hemichannels;(2)They could also be subject to different post-translational modifications, particularly if some kinases are specifically clustered at gap junctional contacts;(3)The docking process between hemichannels when they form gap junctions is likely to induce protein conformational changes, as very large energies are involved in this docking of the extracellular loops that involve many electrostatic and hydrogen-bonding interactions [17]. Certainly, changes would occur in the extracellular domains that mediate docking and also form part of the pore in gap junctions. This region has been shown in structures of Cx46 and 50 gap junctions to contribute significantly to the energy profile of permeant transit [18]. In MD simulations of Cx26 and 30 gap junctions [35], a single charge difference exposed in this region (Lys41 in Cx26 and Glu49 in Cx30) likely explains the differences in ion selectivity of their gap junctions. Propagated conformational changes might also occur within the membrane upon docking of hemichannels to form a gap junction, consistent with the differences seen in the hemichannel structure of Cx31.3 [16] compared with all prior structures of gap junctions [17,18,19].

## 5. Conclusions

The distinctions in the permeation properties of Cx26 gap junctions and hemichannels shown here illustrate that caution is necessary for extrapolating the properties of one to the other. Indeed, there is evidence that the hemichannels comprised of some connexins do not display properties of passive channels at all but act more similarly to selective transporters [12]. This does not appear to be the case for Cx30 or Cx26, but it does indicate that we still have much to learn about the selective permeabilities of connexin channels, not only in terms of how it is affected by the Cx isotype but also by the channel format—hemichannels or gap junctions.

The differences in permeability between Cx30 and 26, the main connexins expressed by supporting cells in the ear, may also help to explain several in vivo studies on hearing in mice. Mutations in connexins, particularly, Cx26, are the most common cause of non-syndromic deafness [36]. In vitro and in vivo experiments in several mouse models, comprehensively reviewed in [37], indicate major roles of Cx26 and 30 in 1—the creation and maintenance of the elevated endolymph potential through K^+^ recycling (possibly via intercellular IP_3_ signaling); 2—the maturation of the Inner Hair Cell (IHC) synapses, linked to extracellular Ca^+2^ wave propagation mediated by ATP release through hemichannels; 3—the maintenance of Outer Hair Cell (OHC) health, likely through the propagation of nutrients/energy resources (glucose or ATP) in the largely avascular cochlear. The relative roles of Cx26 and 30 in the latter two conditions, based on the permeability studies presented here, are illustrated in Figure 7.

Cx26 (blue) and Cx30 (red) hemichannels and gap junctions are shown in cochlear supporting cells, with their permeabilities to ATP quantitively reflected by the width of the green arrows. Both Cx hemichannels have similar permeability to ATP (Cx30 is ~50% larger), which can trigger propagating/regenerative Ca^++^ waves through the P2Y receptor signaling indicated. Intercellular transmission of ATP as a major energy source in an avascular tissue predominantly occurs through Cx26 channels (six-fold more permeable than Cx30 channels).

Altogether, mouse studies have concluded that Cx26 is likely the most important connexin for maintaining hearing. While Cx30 deletion does cause deafness, it also results in loss of Cx26 expression, and if the latter is even partially restored, normal hearing results [38]. This does not explain how many point mutations in Cx30, which presumably do not result in Cx26 reduction, are associated with deafness, but most of these have been associated with excessive hemichannel opening [4] or modified channel permeability [39]. Our studies suggest that both Cx26 and 30 contribute to Ca^++^ wave propagation, and perhaps both are needed to create sufficient ATP flux to mediate normal IHC development. However, in terms of intercellular distribution of energy resources thought to be essential for the maintenance of OHC viability in the absence of vasculature in the cochlear, we show that Cx26 gap junctions are far more effective, consistent with the mouse studies demonstrating a more significant role for Cx26 over Cx30. It would be of interest in the future to also assess the relative glucose permeability of these two channels, the other major source of energy for OHCs. It is also important to note that heteromeric channels of both connexins could have properties that diverge from either Cx alone, and this should be investigated in the future to understand their relative contributions to cochlear function fully.

## Figures and Tables

**Figure 1 life-13-00390-f001:**
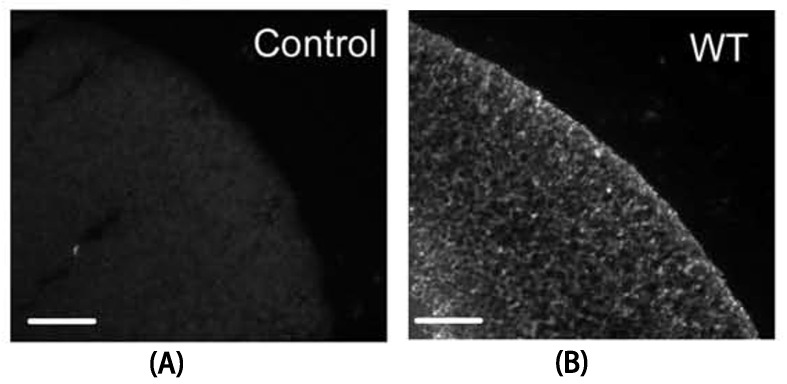
Immunostaining for Cx30 of sections from control oocytes without RNA injection (**A**) and hCx30WT-expressing oocytes (**B**). Regions of the vegetal pole are shown. Scale bars represent 100 µm.

**Figure 2 life-13-00390-f002:**
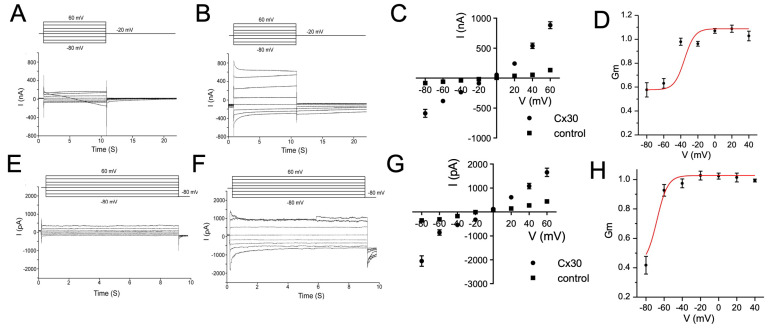
**Cx30 forms hemichannels in *Xenopus oocytes* and N2A cells:** (**A**–**D**): Oocyte expression system; transmembrane currents in response to increasing voltage steps of both polarities in oocytes bathed in Ca^2+^-free ND96 solution injected with antisense oligo alone (**A**) or in conjunction with hCx30 RNA (**B**). Initial current amplitudes plotted against the voltage of Cx30- and antisense-oligo-injected oocytes are linear (**C**)**,** while steady-state conductances plotted against the voltage of Cx30-expressing oocytes (with antisense oligo subtracted) show a partial closure at membrane potentials below −20 mV (Boltzman fit shown in red—V_0_ ~ −35 mV) (**D**). (**E**–**H**): N2A expression system; transmembrane current in response to the increasing voltage steps of both polarities from control (**E**) and hCx30-expressing N2A cells (**F**) bathed in a Ca^2+^-free solution. Initial current amplitude plotted against voltage of transmembrane recordings of control and hCx30-expressing N2A cells are linear, with a slight increase at −80 mV in Cx30-expressing cells (**G**). Steady-state membrane conductance plotted against the voltage of Cx30-expressing N2A cells (conductance from control cells subtracted) showed a decay at membrane potentials below −60 mV (Boltzman fit shown in red) (**H**).

**Figure 3 life-13-00390-f003:**
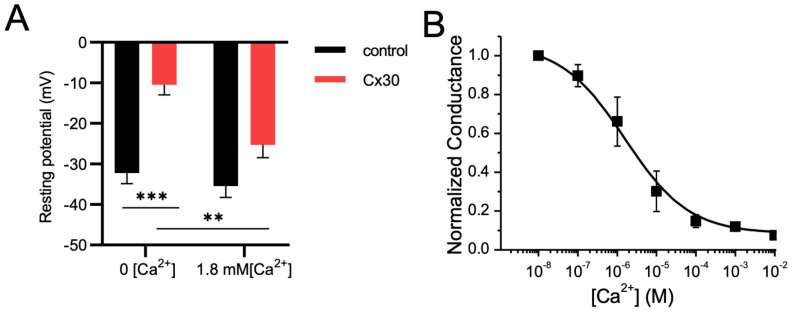
**Cx30 hemichannels sensitivity to extracellular [Ca^2+^]:** (**A**) If Ca^++^ is removed from the media, then Cx30-expressing oocytes show a drop in membrane potential to −10 mV compared with −30 mV in the absence of Cx30 expression, while in the presence of normal extracellular Ca^++^, no significant difference in membrane potential was evident. (**B**) By chelating extracellular Ca^2+^ with EGTA to a series of concentrations, normalized membrane conductance could be plotted against the log of [Ca^2+^]_o_ concentration to measure the Ca^++^ sensitivity of Cx30 hemichannels (Kd of ~ 10 µM). ** *p* < 0.01; *** *p* < 0.001, based on a two-tailed *t*-test.

**Figure 4 life-13-00390-f004:**
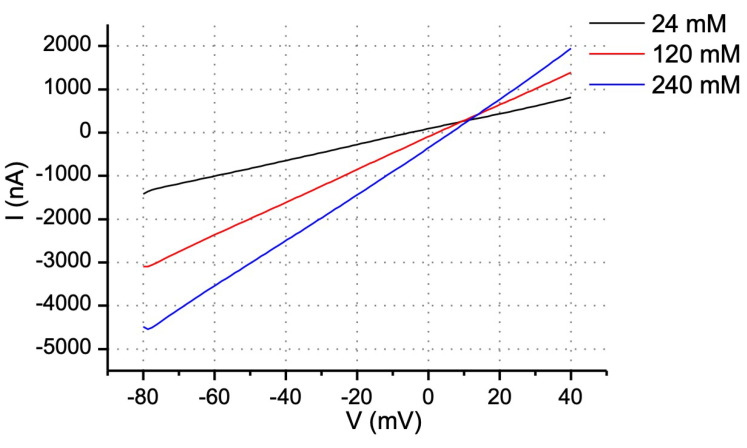
**I–V relations of Cx30 hemichannels in 24, 120 and 240 mM extracellular NaCl.** Current traces of Cx30 hemichannels were recorded in 24 mM (blue), 120 mM (red) and 240 mM (black) extracellular NaCl. Calcium in all the solutions is chelated by EGTA to 10 nM. Note that the intersection point of the currents is in the first quadrant and that reversal potentials shift to positive values with increasing NaCl concentration, allowing calculations of relative Na^+^, K^+^ and Cl^−^ conductances by the GHK equation.

**Figure 5 life-13-00390-f005:**
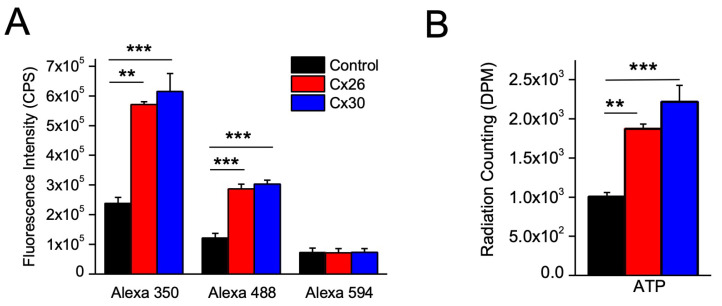
**Permeability of Cx30 hemichannel to Dyes and ATP.** (**A**) Three Alexa dyes of increasing size (350 (MW 350), 488 (MW 570) and 594 (MW 760)) were injected into oocytes previously injected with antisense oligo to endogenous XeCx38 alone (white) or co-injected with cRNA for Cx26 (grey) or Cx30 (black). Fluorescence intensity was released following incubation in Ca^2+^-free KCl solution, which was measured at 60 min. (**B**) Precisely 32 nL of 1.25 mCi/mL ^35^S-labeled ATP-γ-S was injected into oocytes as described in (**A**). Radiation released following incubation in Ca^2+^-free KCl solution was measured at 60 min. Data are means ± SEM; *n* = 5; ** *p* < 0.01; *** *p* < 0.001, based on a two-tailed *t*-test.

**Figure 6 life-13-00390-f006:**
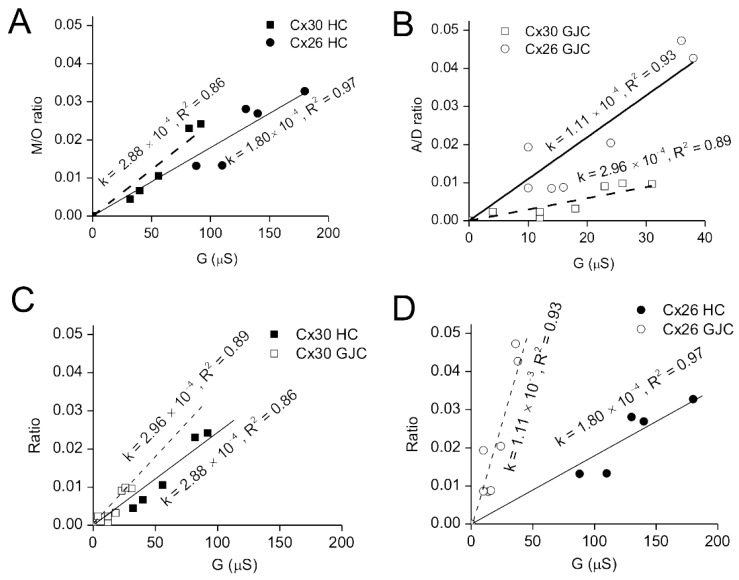
Comparison of Cx26 and Cx30 hemichannel and gap junction channel permeability to ATP-γ-S. (**A**) Linear fit of the ratio of released ATP-γ-S to the medium (M) to ATP-γ-S retained in the injected oocyte (O) against electrical conductance, measured by a two-electrode voltage clamp prior to injection. The average value of released ATP-γ-S and electrical conductance from control oocytes (not expressing connexins) is subtracted from each measurement. (**B**) Linear fit of ATP-γ-S transferred from an injected donor (**D**) to acceptor (**A**) oocyte through gap junction channels (presented as A/D ratio) against transjunctional conductance immediately measured prior to ATP injection into the donor cell. The average value of ATP-γ-S transfer and electrical conductance from control oocytes (not expressing connexins) is subtracted from each measurement. (**C**,**D**) Since the recordings are collected under identical conditions in terms of ATP injection and time of transfer, the plots in A and B can be combined on the same axes to provide a direct comparison of hemichannel and gap junction channel permeabilities for Cx30 (**C**) and Cx26 (**D**). In all cases, ATP injection and collection were performed as described in Figure 4. For the gap junction assays, the acceptor and donor oocytes are separated and processed for counting after 1 h.

**Figure 7 life-13-00390-f007:**
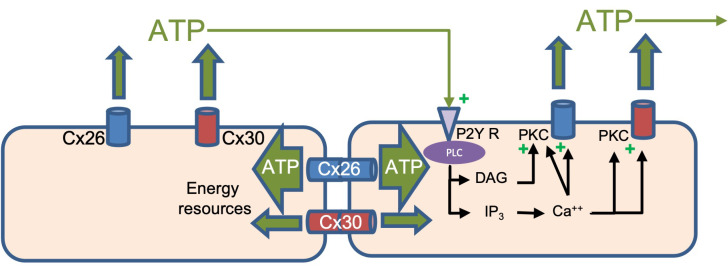
Model of the physiological effects of differential permeabilities of Cx26 and 30 hemichannels and gap junctions.

**Table 1 life-13-00390-t001:** Effect of extracellular ion concentration on reversal potential.

NaCl Concentration (mM)	RP (mV)
24	−8.2 ± 0.7
120	3.2 ± 0.4
240	10 ± 2.5

**Table 2 life-13-00390-t002:** Reversal potential shifts (mV) of Cx30 hemichannel current induced by substitution of extracellular Na^+^ or Cl^−^ with organic ions.

% Substitution	TEA^+^ (MW: 130)	Gluconate^−^ (MW: 196)
0	−0.3 ± 0.7	−0.2 ± 1.8
50	−6.6 ± 0.4	−2.5 ± 0.7
100	−15.3 ± 2.5	−5.1 ± 0.1

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
