# Peer review of "Divergence between Hemichannel and Gap Junction Permeabilities of Connexin 30 and 26"

_life, 2023, doi:10.3390/life13020390_

Round 1
Reviewer 1 Report
Connexin 26 and 30 are members of the connexin protein family. Mutations in the connexins in the cochlear epithelium, Cx26 and Cx30, cause sensorineural deafness. Connexins function as gap junction channels and as hemichannels. In the manuscript titled “Divergence between hemichannel and gap junction permeabilities of Connexin 30 and 26” by Xu et al., the authors used heterologous expression systems to examine the properties of Cx26 and Cx30. The authors demonstrated differences in Cx30 and Cx26 hemichannel permeabilities. However, a few control experiments are missing, and the formation of tight junctions is not examined in their system. I have detailed my comments below.
1. Control group is missing in a few experiments. The authors should include oligo-injected oocytes in Figure 5.
2. The authors should show the formation of Cx26 hemichannels in Xenopus oocytes and N2A cells similar to Figure 1 and 2.
3. Overexpression of Cx30 and Cx26 should be validated by western blot or immunohistochemistry.
4. In the case of examining gap junction permeabilities of Cx26 and Cx30 in Figure 5, the authors should use immunohistochemistry to show that gap junctions were indeed formed between cells. The authors can use injected fluorescent dye to validate the transport between cells.
Author Response
Connexin 26 and 30 are members of the connexin protein family. Mutations in the connexins in the cochlear epithelium, Cx26 and Cx30, cause sensorineural deafness. Connexins function as gap junction channels and as hemichannels. In the manuscript titled “Divergence between hemichannel and gap junction permeabilities of Connexin 30 and 26” by Xu et al., the authors used heterologous expression systems to examine the properties of Cx26 and Cx30. The authors demonstrated differences in Cx30 and Cx26 hemichannel permeabilities. However, a few control experiments are missing, and the formation of tight junctions is not examined in their system. I have detailed my comments below.
- Control group is missing in a few experiments. The authors should include oligo-injected oocytes in Figure 5.
We agree that this is a critical control. The Oligo injected controls are not plotted separately, as these values were subtracted from all data sets in Fig. 5, as indicated in Methods and the Figure legend. Unlike Histograms, it was much more instructive to subtract the background from each data point before plotting, so only values above this background are shown
- The authors should show the formation of Cx26 hemichannels in Xenopus oocytes and N2A cells similar to Figure 1 and 2.
Given the extensive prior characterization of Cx26, the focus of this manuscript in on Cx30 gap junctions and hemichannels. However, to document expression, we now include gap junction records from Cx26 injected oocytes in Fig. S2.
- Overexpression of Cx30 and Cx26 should be validated by western blot or immunohistochemistry.
As we show records of functional expression of both Cx26 and 30, with properties consistent with prior reports on these two connexins, we felt that additional demonstration of protein expression was not needed. However, we have now included immunofluorescent images of control and Cx30 injected oocytes in Fig. S1 as an example demonstrating protein expression.
- In the case of examining gap junction permeabilities of Cx26 and Cx30 in Figure 5, the authors should use immunohistochemistry to show that gap junctions were indeed formed between cells. The authors can use injected fluorescent dye to validate the transport between cells
We have used functional measurements, rather than immunofluorescence, to validate the formation of gap junctions and hemichannels in both oocyte and N2A cell systems. In each case, both conductances, as well as dye or ATP leak or intercellular transfer were markedly greater in Cx26 and 30 expressing cells than control cells. In Fig. 5 specifically, we normalized ATP leak to membrane conductance, and ATP intercellular transfer to intercellular conductance, with the background conductances from uninjected cells subtracted in all cases. We believe this is a more accurate measure of the number of hemichannels or gap junction channels connecting two cells, respectively, than is dye transfer, as the latter is influenced more by differential selectivity of the two pores. We have previously published an extensive analysis establishing the linear relationship between intercellular conductances and dye transfer for several connexins that validate this approach (Weber et al. 2004), but also show it can vary between connexins, which is what we again demonstrate here. However, we extend these observations to show that it can also vary between hemichannel and gap junction channels comprised of the same connexin.

Reviewer 2 Report
The manuscript has been carefully evaluated. The authors deal with an interesting topic. However, major modifications are needed. Please note my comments below:
Please chnage the references into numbers, and the reference list should be according to the appearance in the text.
Please and microscopic pictures of the oocytes, before and after staining.
The general quality of the graphs appears to be very poor, please fix!
Figure 4: Please add a picture of the oocyte with regards to the used dyes.
Although there is a concluding sentence at the end of the discussion, a conclusion section should be added.
Author Response
The manuscript has been carefully evaluated. The authors deal with an interesting topic. However, major modifications are needed. Please note my comments below:
Please change the references into numbers, and the reference list should be according to the appearance in the text.
This has now been done to fit Journal format.
Please and microscopic pictures of the oocytes, before and after staining.
As an example, we now include pictures of control and Cx30 expressing oocytes in Fig. S1. We do NOT see any phenotypic changes in Cx30 or 26 expressing oocytes or N2A cells, such as is often seen when injecting mutant connexins that forms too many open hemichannels.
The general quality of the graphs appears to be very poor, please fix!
Resolution of some images was non-optimal due to their insertion in the text, while keeping the total file size below Journal requested limits. Better resolution, and color images where requested, are now provided in a separate PDF.
Figure 4: Please add a picture of the oocyte with regards to the used dyes.
All dye leak and transfer was measured quantitively by fluorimeter, so no images were recorded. Images were previously reported extensively in Weber at al., 2004.
Although there is a concluding sentence at the end of the discussion, a conclusion section should be added.
We thank the reviewer for this excellent suggestion, and have now included both a Conclusion section, as well as Fig. 6 illustrating the major points. This has allowed us to provide better physiological context for the study.
Reviewer 3 Report
This is a good article, but it still needs some improvement.
1) Figure 1 should be statistically processed
2) Figure 4 should be colored
3) It would be nice to present a photo of the object of research
4) No conclusions section
Author Response
This is a good article, but it still needs some improvement.
- Figure 1 should be statistically processed
We thank the reviewer for this suggestion, and have now included plots of our compiled current records, as well as a Boltzman plot of conductances against voltage as additional panels in Fig. 1 (C,G and D,H, respectively). Note that plots at higher positive potentials were not possible due to a large endogenous outward current that was activated, particularly in oocytes. In addition, in Fug. 2, we have now included compiled data, with statistical analysis of the effect of low Ca++ on membrane conductance in control and Cx30 injected oocytes (this data has been previously published for Cx26).
2) Figure 4 should be colored
Thank you for this suggestion, as it makes the graphs more readable, and has also been employed in the graph in Fig. 2.
- It would be nice to present a photo of the object of research
We have now included a Fig 6, which presents a graphical summary of the conclusions of our study and relates to their physiological relevance that is now presented in a new Conclusions section
4) No conclusions section
See above
Round 2
Reviewer 1 Report
The authors have addressed most of my concerns in the revised version of the manuscript. I recommend the paper for publication.
Author Response
Thank you for your diligent review...it improved the final manuscript
Reviewer 2 Report
Please add the the oocyte pictures to the main Manuscript, since these are crucial and important information for the reader.
Author Response
The oocytes pictures are now added as Fig. 1, and all Fig numbering has been corrected....thank you. I left one supplementary Figure (Cx26 HC traces), as it does not fit well in the flow of the manuscript and its figures
Reviewer 3 Report
My comments have been corrected and the article can be accepted for publication.
Author Response
Thank you...your constructive critique has helped improve teh final manuscript